# Influence of Genetic Variations in miRNA and Genes Encoding Proteins in the miRNA Synthesis Complex on Toxicity of the Treatment of Pediatric B-Cell ALL in the Brazilian Amazon

**DOI:** 10.3390/ijms24054431

**Published:** 2023-02-23

**Authors:** Elisa da Silva Menezes, Francisco Cezar Aquino de Moraes, Amanda de Nazaré Cohen-Paes, Alayde Vieira Wanderley, Esdras Edgar Batista Pereira, Lucas Favacho Pastana, Antônio André Conde Modesto, Paulo Pimentel de Assumpção, Rommel Mario Rodríguez Burbano, Sidney Emanuel Batista dos Santos, Ney Pereira Carneiro dos Santos, Marianne Rodrigues Fernandes

**Affiliations:** 1Oncology Research Center, Federal University of Pará, Belém 66073-005, PA, Brazil; 2Otávio Lobo Children’s Cancer Hospital, Belém 66063-005, PA, Brazil; 3State Department of Public Health (SESPA), Belém 66093-677, PA, Brazil; 4Laboratory of Human and Medical Genetics, Institute of Biological Science, Federal University of Pará, Belém 66077-830, PA, Brazil; 5Instituto Tocantinense Presidente Antônio Carlos (ITPAC), Abaetetuba 68440-000, PA, Brazil; 6Ophir Loyola Hospital, Molecular Biology Laboratory, Belém 66063-240, PA, Brazil

**Keywords:** acute lymphoblastic leukemia, toxicity, miRNA, ancestry, SNVs

## Abstract

Acute lymphoblastic leukemia (ALL) is the most common childhood cancer in the world. Single nucleotide variants (SNVs) in miRNA and genes encoding proteins of the miRNA synthesis complex (SC) may affect the processing of drugs used in the treatment of ALL, resulting in treatment-related toxicities (TRTs). We investigated the role of 25 SNVs in microRNA genes and genes encoding proteins of the miRNA SC, in 77 patients treated for ALL-B from the Brazilian Amazon. The 25 SNVs were investigated using the TaqMan^®^ OpenArray™ Genotyping System. SNVs rs2292832 (*MIR149*), rs2043556 (*MIR605*), and rs10505168 (*MIR2053*) were associated with an increased risk of developing Neurological Toxicity, while rs2505901 (*MIR938*) was associated with protection from this toxicity. *MIR2053* (rs10505168) and *MIR323B* (rs56103835) were associated with protection from gastrointestinal toxicity, while *DROSHA* (rs639174) increased the risk of development. The rs2043556 (*MIR605*) variant was related to protection from infectious toxicity. SNVs rs12904 (*MIR200C*), rs3746444 (*MIR499A*), and rs10739971 (*MIRLET7A1*) were associated with a lower risk for severe hematologic toxicity during ALL treatment. These findings reveal the potential for the use of these genetic variants to understand the development of toxicities related to the treatment of ALL in patients from the Brazilian Amazon region.

## 1. Introduction

Acute lymphoblastic leukemia (ALL) is the most common childhood cancer, and B-cell acute lymphoblastic leukemia (B-ALL) accounts for 75–80% of all ALL cases [1]. This leukemic type is the cancer with the highest lethality from 0 to 19 years of age in Brazil [2]. Although the survival rate recorded in pediatric patients with ALL in developed countries such as Australia and Belgium is >90%, the treatment of this neoplasm is still challenging as serious toxic events continue to be present in about 20% of children treated for ALL [3,4].

Pharmacological therapy for ALL uses the European Group Berlin–Frankfurt–Münster (BFM) as its main protocol, which consists of a combination of chemotherapeutic agents according to the stage of treatment [1]. These chemotherapeutics have a narrow therapeutic window, which favors high toxicity in children treated for ALL, often generating changes or the interruption of therapy [4,5]. The response to treatment varies inter-individually among patients with the same protocol; this is partly due to pharmacogenomic variability, as already well established and recommended by international agencies (FDA and EMA) for the *TPMT* and *NUDT15* genes [6].

The literature demonstrates that some factors influence the development of treatment-related toxicities (TRTs), including the patient’s ethnic origin, since the peculiarity of genetic signatures of mixed populations in the Brazilian Amazon, with different degrees of ancestry and with greater contribution of Amerindian descent, plays a significant role in modulating ALL chemotherapy susceptibility and toxicity [7].

Likewise, single nucleotide variants (SNVs) present in miRNA genes and in genes encoding proteins of the miRNA synthesis complex may affect the function of drugs used for treating ALL and contribute to the efficacy and development of toxicity [8]. Pharmacogenomics studies of ALL with mixed populations from the Brazilian Amazon have been published in the scientific literature, highlighting the significance of the regulatory role of genetic variants involved in the development of ALL and the impact of toxicities during pharmacological therapy in pediatric patients [4,7].

Therefore, our study investigated the role of 25 SNVs located in important miRNA genes and in genes encoding proteins of the miRNA synthesis complex, with the toxicity of pediatric patients from the Brazilian Amazon region treated for B-cell ALL.

## 2. Results

The clinical data of the 77 patients with ALL included in this study are presented in Table 1. The mean age of the patients was 4.8 ± 3.6 years, with the majority being male with 62.3%. Study participants had an average proportion of European ancestry of 42.7%, followed by Amerindians with 36.3% and Africans with 20.4%.

Regarding the toxicity parameters, among the 77 patients, 90.9% had some type of toxicity, of which 78.5% had severe toxicities, with the most frequent being infectious toxicity (85.7%), followed by gastrointestinal toxicity present in 67.1% of patients, in addition to hematological (62.8%) and neurological (22.8%) toxicity. Because of important toxicity findings, we stratified the sample according to each outcome (Table 1).

In the neurological-type toxicity results, four genetic variants located in *MIR149*, *MIR605*, *SVIL*, and *MIR2053* showed statistical relevance with significant results (Table 2). The rs2292832 mutant homozygous TT genotype (*MIR149*) was associated with a 7-fold increased risk of developing neurological toxicity compared with patients with other genotypes (*p* = 0.016, OR = 7.26). The homozygous mutant TT genotype of rs2043556 (*MIR605*) also presented results with an association of risk in the development of neurological toxicity during the treatment of ALL, suggesting a 10-fold increase in relation to the other genotypes (*p* = 0.039, OR = 10.23). Thus, the homozygous mutant TT genotype of rs10505168 (*MIR2053*) was associated with a fourfold increased risk of developing neurological toxicity compared with the other genotypes (*p* = 0.025, OR = 4.61). In contrast, the homozygous mutant TT genotype of rs2505901 (*SVIL*) was significantly associated with a lower risk of developing neurological toxicity, suggesting a protective factor for patients with this genotype (*p* = 0.030, OR = 0.09).

In the specific analyses for gastrointestinal toxicity, three polymorphisms with a significant association were identified (Table 3). The rs2505901 (*SVIL*) polymorphism was shown to be a protective factor for individuals with the CC genotype compared with patients with other genotypes (*p* = 0.045, OR = 0.20). The homozygous mutant rs639174 (CC) genotype in the *DROSHA* gene was significantly associated with an 11-fold increased risk of developing gastrointestinal toxicity during ALL treatment (*p* = 0.040, OR = 11.63). The wild-type homozygous TT variant of rs56103835 (*MIR323B*) was shown to be a protective factor for gastrointestinal toxicity in patients with this genotype compared with the others in this study. None of the genetic variants investigated demonstrated significance for levels of severe gastrointestinal toxicity. The logistic regression performed was adjusted for African ancestry (*p* = 0.026).

One polymorphism showed a statistically significant result with infectious-type toxicity during ALL treatment (Table 4): rs2043556 of the *MIR605* gene. The homozygous wild-type (CC) genotype of rs2043556 was associated with a reduced risk of developing infectious toxicity (*p* = 0.010, OR = 0.08), suggesting a protective effect.

In Table 5, analyses of severe hematological toxicity revealed significant results for three variants: rs12904 of the *MIR200C* gene (*p* = 0.037, OR = 0.26), rs3746444 of the *MIR499A* gene (*p* = 0.025, OR = 0.23), and rs10739971 of the *MIRLET7A1* gene (*p* = 0.040, OR = 0.18). Both variants were associated with a lower risk of developing severe hematologic toxicity during ALL treatment.

## 3. Discussion

Despite the clinical advances in the treatment of ALL, 65.3% of the mixed population with a high contribution of Amerindian ancestry still present grades 3 and 4 toxicities when submitted to the European Group Berlin–Frankfurt–Münster (BFM) protocol, data that present great difference when compared with the average of world populations (20%) [4,10]. Due to the high rate of miscegenation in poorly investigated populations, such as Brazilian, it is important to evaluate genetic variants that may be related to therapeutic failure. Some studies have sought to identify genetic variants that may be directly related to the high toxicities during the treatment of ALL found in the Brazilian Amazon; in this way, the present project intends to evaluate the effect of important genetic variants in miRNA genes for the toxicities generated in Amazonian populations [10].

In this study, significant relationships were found with the following variants: rs2292832 (*MIR149*), rs2043556 (*MIR605*), rs10505168 (*MIR2053*), and rs2505901 (*MIR938*), correlated with neurological toxicity during the ALL treatment. *MIR149* has previously been linked to the risk of multiple cancers, including colorectal, liver, and breast cancer [11,12,13]. In our study, rs2292832 (*MIR149*) was statistically associated with an increased risk of central nervous system (CNS) toxic events during ALL treatment.

The rs2043556 (*MIR605*) variant has also been linked to the development of a variety of cancers, including breast and gastrointestinal cancer; it may affect the functionality of the *MIR605* processing gene [14,15]. The *MDM2* gene is involved in different parts of the cancer signaling cascade, being a direct target of regulation by MIR605 [14,15]. This miRNA is considered an activator of the p53 signaling pathway, interrupting the p53-MDM2 interaction, resulting in an accumulation of p53 and consequently aiding its cellular functionality [15,16]. The G allele of rs2043556 was associated with its interference in platinum-based therapy, inducing the risk of severe hepatotoxicity in lung cancer patients during platinum-based treatment [17].

In this investigation, the *MIR605* gene rs2043556 was significantly associated with the risk of developing neurological toxicity inherent to the use of MTX and 6-Mercaptopurine and also associated with a 92% protective factor in the risk of infectious toxicities in our patients. This variant has also been previously evidenced as a phenotype modeler in patients with Li-Fraumeni syndrome [18].

In this study, the rs10505168 of the *MIR2053* gene was associated with the development of neurological toxicities during the treatment of ALL. This variant was identified and related to oral mucositis in the consolidation stage of ALL treatment [19]. In addition, another study analyzed the association of this variant with the risk of MTX-induced oral mucositis in Dutch children with ALL [20].

Statistically significant results were found in *DROSHA* (639174), *MIR198* (2505901), and *MIR323B* (56103835), with the risk of developing gastrointestinal toxicities in our patients. *DROSHA* encodes double-stranded RNA-specific ribonuclease III (RNase III), an important enzyme for the production of pre-miRNA from pri-miRNAs. SNVs in this gene can cause changes in drug responses as a result of their expression variation. The first study to demonstrate the potential of variants in miRNA processing genes as a predictor of toxicity in the pharmacological management of ALL linked rs639174 in *DROSHA* to MTX-induced gastrointestinal toxicity in pediatric patients with B-cell ALL [19,21]. In our study, this result was confirmed, and the rs639174 variant was associated with an increased risk of developing gastrointestinal toxic effects.

The rs56103835 variant in pre-*MIR323B* was previously associated with changes in plasma MTX levels and the occurrence of vomiting in patients with ALL; mature *MIR453* may undergo alterations in its levels and biogenesis and, consequently, in its role in the regulation of target genes *ABCC1, ABCB1, ABCC2,* and *ABCC4*, included in the transport of the drug MTX. The rs56103835 variant was related to mature MIR functioning and plasma levels of this drug during pharmacological treatment, which supports a significant association with gastrointestinal toxicity [19]. The results of the present study indicate that the rs56103835 variant was associated with a lower risk of developing gastrointestinal toxicities.

In our study, the rs2505901 variant of the *MIR938* gene was significantly associated with protection from gastrointestinal and neurological toxicity. Another variant present in the *MIR938* gene has been associated with decreased risk of gastric cancer [22]. In addition, rs2505901 has been previously described as a potential reduction in ALL susceptibility in populations from the Amazon region, which corroborates the protective effect of the rs2505901 variant [23].

Regarding severe hematologic toxicity, our results demonstrated significant data on the variants rs12904 (*MIR200C*), rs3746444 (*MIR499A*), and rs10739971 (*MIRLET7A1*) for hematologic toxicity arising from the treatment of ALL in our patients. In the present work, rs12904 was associated with a 74% decrease in the risk of hematologic toxicity. The same variant was previously related to colorectal cancer risk in the Chinese population and gastric adenocarcinoma risk [24,25]. Studies showed that MIR200C was expressed at significantly lower levels in children with relapsing ALL compared to those at primary diagnosis, i.e., the decreased expression of MIR200C can be considered as a prognostic factor for relapse in childhood ALL [26].

Another significant association for hematologic toxicity in our study involved rs3746444 (*MIR499A*). This variant was identified and associated with a 77% lower chance of developing toxic hematological events during the treatment of ALL in the present study. rs3746444 was recently implicated in ALL susceptibility in a study that relevantly associated this variant with a lower risk of developing ALL [20], while another demonstrated an association of rs3746444 with significantly increased risk of ALL [27].

In our study, the rs10739971 variant of the *MIRLET7A1* gene was associated with a possible protective factor of 82% in the risk of developing toxic hematological events in pediatric patients undergoing ALL. The same variant has already been described as a potential gastric cancer biomarker [28].

Three variants in our study (rs10505168, rs639174, and rs56103835) were previously associated and described in the specialized literature as relevant to the pharmacogenomics of ALL treatment [19]. In addition, other new variants were identified and related for the first time to toxic events during chemotherapy for ALL.

Therefore, we suggest that variants in the *MIR149, MIR605, MIR938, MIR200C, MIR499A,* and *MIRLET7A1* genes, although not related in the specialized literature to the pharmacological pathways of the antineoplastics used in ALL, can still influence the toxicity in these patients since they play regulatory roles in the cellular environment aiding the survival of cancer cells, thus impairing the effectiveness of the treatment.

It is important to highlight that the expression of variants in miRNAs can affect the function of drugs; however, they are still poorly understood in the involvement of different pathways that may be regulating the treatment of ALL [8]. Therefore, additional studies would be ideal to clarify the real potential of these variants in the pharmacological management of ALL.

## 4. Materials and Methods

### 4.1. Ethics, Consent, and Permissions

All precepts of the Declaration of Helsinki and the Nuremberg Code were followed, in addition to the Research Standards Involving Human Beings in Brazil (Res. 466/12 of the National Health Council). The study protocol was approved by the Research Ethics Committee of the Research Center of Oncology of the Federal University of Pará (CAAE number 11433019.5.0000.5634/2019).

### 4.2. Investigated Population

The study was performed with 77 patients diagnosed with B-cell ALL through immunophenotyping and/or molecular analysis, aged between 1 and 18 years. All of them received treatment at the Otávio Lobo Children’s Oncological Hospital, a reference center for childhood cancer treatment in the North region of Brazil. Patients with a history of relapses or with comorbidities were excluded from this study. Toxicity data were collected from patient charts and were classified based on common terminology criteria for adverse events v.5 [9].

### 4.3. Selected Bookmarks

The SNVs selected for the present study were chosen due to their participation in the pharmacokinetics and/or pharmacogenomics of one or both drugs (6-MP or MTX) used in the treatment of ALL and associated with adverse events, in addition to being involved in this carcinogenesis. A total of 25 variants were selected in miRNA genes and in genes encoding proteins essential for the synthesis of miRNAs contained in Appendix A. The selection was performed based on the inclusion of two criteria: (i) MAF ≥ 1% and (ii) genotyping rate ≥80.

### 4.4. DNA Extraction and Quantification

DNA was extracted from peripheral blood using the commercial BiopurKit Mini Spin Plus–250 Extraction Kit (Biopur, Pinhais, PR, Brazil), according to the manufacturer’s instructions. The concentration of genetic material was quantified using a NanoDrop 1000 spectrophotometer (NanoDrop Technologies, Wilmington, DE, USA).

### 4.5. Genotyping

Genotyping of variants was performed by allelic discrimination using QuantStudio TaqMan^®^ OpenArray technology (ThermoFisher, Carlsbad, CA, USA) with a set of 25 custom assays, which were run on a QuantStudio™ 12K Flex Real-Time PCR system (Applied Biosystem, Life Technologies, Carlsbad, CA, USA), according to the manufacturer’s protocol. The quality of genotype readings and other data were analyzed using the TaqMan^®^ v1.2 Genotyper software (Thermo Fisher Scientific).

### 4.6. Genetic Ancestry

The genomic ancestry of the participants was analyzed according to Santos et al. 2010) [29] and Ramos et al., 2016 [30], using a set of 61 Ancestry Informative Markers (AIMs). The individual and global proportions of European, Amerindian, and African genetic ancestry were estimated using STRUCTURE v.2.3.4.

### 4.7. Statistical Analysis

A descriptive analysis of the data referring to the characterization of the sample was carried out using the absolute frequency; percentage; mean; standard deviation; median; and interquartile range of 25% to 75%. Quantitative variables were first submitted to the Kolmogorov–Smirnvov test for normality distribution analysis. The individual proportions of European, African, and Amerindian genetic ancestry were estimated using Structure 2.3.3 software.

The Chi-square test was applied to categorical variables and the Mann–Whitney test to continuous variables. To analyze the association of polymorphisms involved with the risk of toxicity in ALL, simple and multivariate logistic regression was performed. All statistical analyzes were performed using the SPSS 20.0 statistical package, considering a significance level of 5% (*p*-value ≤ 0.05).

## 5. Conclusions

In the mixed population of the North region of Brazil, the average occurrence of toxicities is above the average of the rest of the population, and in part, the specialized literature has demonstrated the influence of Amerindian ancestry. Few studies describe data on miRNA variants and their influence on ALL therapy in mixed Brazilian populations. Therefore, the data presented are of great value for understanding the variability of response in the standard treatment of ALL in the investigated population, standing out among the other studies carried out in homogeneous populations, which were the focus of most previous research on this topic. Finally, we demonstrate that the investigated *MIR149*, *MIR605*, *MIR938*, *DROSHA*, *MIR200C*, *MIR499A*, *MIRLET7A1*, *MIR323B*, and *MIR2053* could potentially modulate the therapeutic response of standard treatment for ALL.

## Figures and Tables

**Table 1 ijms-24-04431-t001:** Clinical characteristics of the patients analyzed in the present study.

Variables	Frequency (%)	CI 95%
**Gender**		
Female	29 (37.7)	27.3–49.3
Male	48 (62.3)	50.7–72.7
**Age**		
Average	4.8 (±3.6) ^1^	4.1–5.6
Median	4.0 (2.0–6.0) ^2^	3.0–4.0
**Ancestry**		
European	42.7 * (±13.7) ^1^	39.5–45.9
Amerindian	36.3 * (±15.5) ^1^	32.7–39.9
African	20.4 * (±8.9) ^1^	18.4–22.6
**General Toxicity**		
Yes	70 (90.9)	84.4–97.4
No	7 (9.1)	2.6–15.6
**Severe Toxicity ^#^**		
Yes	55 (78.5)	68.6–88.6
No	15 (21.4)	11.4–31.4
**Types of Toxicity ^#^**		
Infectious	60 (85.7)	70.7–88.0
Gastrointestinal	47 (67.1)	55.8–77.1
Hematologic	44 (62.8)	51.4–74.3
Neurologic	16 (22.9)	14.3–32.9
**Mortality**		
Yes	31 (40.3)	28.6–51.9
No	46 (59.7)	48.1–71.4

* Median; ^1^ standard deviation value; um ^2^ percentile (25–75%); ^#^ toxicity classified according to common terminology criteria for adverse events v.5 [9].

**Table 2 ijms-24-04431-t002:** Analysis of miRNA polymorphisms for neurological toxicity.

ID SNV	Neurological Toxicity		*p*-Value ^a^	OR (CI 95%) ^b^
	Yes	No		
*MIR149*_rs2292832				
CC	0 (0.0)	1 (2.4)	**0.016**	TT vs. others
CT	2 (14.3)	22 (52.4)		
TT	12 (85.7)	19 (45.2)		7.26 (1.44–36.54)
Allele C	2 (7.1)	24 (28.6)		
Allele T	26 (92.9)	60 (71.4)		
*MIR605*_rs2043556				
CC	0 (0.0)	8 (25.0)	**0.039**	TT vs. others
CT	1 (12.5)	11 (34.4)		
TT	7 (87.5)	13 (40.6)		10.23 (1.12–93.34)
Allele C	1 (6.2)	27 (42.2)		
Allele T	15 (93.8)	37 (57.8)		
*MIR938_rs2505901*				
CC	4 (26.7)	7 (16.7)	**0.030**	TT vs. others
CT	10 (66.7)	17 (40.5)		
TT	1 (6.7)	18 (42.9)		0.09 (0.01–0.79)
Allele C	18 (60.0)	31 (36.9)		
Allele T	12 (40.0)	53 (63.1)		
*MIR2053*_rs10505168				
CC	0 (0.0)	4 (21.1)	**0.025**	TT vs. others
CT	4 (28.6)	2 (10.5)		
TT	10 (71.4)	13 (68.4)		4.61 (1.21–17.65)
Allele C	4 (14.3)	28 (37.8)		
Allele T	24 (85.7)	46 (62.2)		

^a^*p*-value < 0.05; ^b^ simple logistic regression.

**Table 3 ijms-24-04431-t003:** Analysis of miRNA polymorphisms for gastrointestinal toxicity.

ID SNV	Gastrointestinal Toxicity		*p*-Value ^a^	OR (CI 95%) ^b^
	Yes	No		
*MIR938*_rs2505901				
CC	5 (13.2)	6 (30.0)	**0.045**	CC vs. others
CT	24 (63.2)	4 (20.0)		
TT	9 (23.7)	10 (50.0)		0.20 (0.04–0.96)
Allele C	34 (44.7)	16 (40.0)		
Allele T	42 (55.3)	24 (60.0)		
*DROSHA*_rs639174				
CC	13 (72.2)	2 (25.0)	**0.040**	CC vs. others
CT	5 (27.8)	5 (62.5)		
TT	0 (0.0)	1 (12.5)		11.63 (1.11–121.55)
Allele C	31 (86.1)	9 (56.3)		
Allele T	5 (13.9)	7 (43.8)		
*MIR323B*_rs56103835				
CC	3 (8.6)	0 (0.0)	**0.045**	TT vs. others
CT	21 (60.0)	6 (37.5)		
TT	11 (31.4)	10 (62.5)		0.23 (0.05–0.96)
Allele C	27 (38.6)	6 (18.3)		
Allele T	43 (61.4)	26 (81.3)		

^a^*p*-value < 0.05; ^b^ Logistic regression adjusted for African ancestry.

**Table 4 ijms-24-04431-t004:** Analysis of miRNA polymorphisms for infectious toxicity.

ID SNV	Infectious Toxicity		*p*-Value ^a^	OR (CI 95%) ^b^
	Yes	No		
*MIR605*_rs2043556				
CC	3 (9.7)	4 (57.1)	**0.010**	CC vs. others
CT	10 (32.3)	2 (28.6)		
TT	18 (58.1)	1 (14.3)		0.08 (0.01–0.54)
Allele C	16 (25.8)	10 (71.4)		
Allele T	46 (74.2)	4 (28.6)		

^a^*p*-value < 0.05; ^b^ Simple logistic regression.

**Table 5 ijms-24-04431-t005:** Analysis of microRNA polymorphisms for severe hematologic toxicity.

ID SNV	Severe Hematologic Toxicity		*p*-Value ^a^	OR (CI 95%) ^b^
	Yes	No		
*EFNA1*_rs12904				
AA	4 (15.4)	14 (41.2)	**0.037**	AA vs. others
AG	16 (61.5)	13 (38.2)		
GG	6 (23.1)	7 (20.6)		0.26 (0.73–0.92)
*MIR499A*_rs3746444				
AA	4 (16.0)	1 (3.2)	**0.025**	GG vs. others
AG	17 (68.0)	16 (51.6)		
GG	4 (16.0)	14 (45.2)		0.23 (0.06–0.83)
*MIRLET7A1*_rs10739971				
AA	10 (52.6)	15 (41.7)	**0.040**	GG vs. others
AG	7 (36.8)	7 (19.4)		
GG	2 (10.5)	14 (38.9)		0.18 (0.04–0.93)

^a^*p*-value < 0.05; ^b^ Simple logistic regression.

## Data Availability

Not applicable.

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
