# Peer review of "Influence of Genetic Variations in miRNA and Genes Encoding Proteins in the miRNA Synthesis Complex on Toxicity of the Treatment of Pediatric B-Cell ALL in the Brazilian Amazon"

_ijms, 2023, doi:10.3390/ijms24054431_

Round 1
Reviewer 1 Report
In this manuscript, the authors evaluated 25 single nucleotide variants (SNVs) of microRNAs and genes in patients with B-cell ALL, and analyzed the effects of these variants on chemotherapy-related toxicities. Although the authors identified some SNVs that are associated with different types of toxicities, there are some critical issues need to be addressed.
1. How did the authors select the 25 SNVs, as some SNVs have never been reported in terms of leukemia. In addition, some SNVs, such as ATXN1 (rs68082256) that was linked with neurotoxicity in pediatric ALL, and SLC19A1 (rs1051266) that was significantly associated with MTX-related toxicity in pediatric ALL, were not on the list. In this regard, the rationale for the 25-SNV testing list is unclear. The selection criteria and supporting references need to be further provided.
2. How did the authors define toxicities? The clinical symptoms and signs or the drug plasma concentrations for each type of toxicity (infectious, gastrointestinal, hematologic and neurologic) need to be clarified in detail.
Author Response
Point 1: How did the authors select the 25 SNVs, as some SNVs have never been reported in terms of leukemia.
Response 1: We appreciate the inquiry. This investigation has some singularities, as we approach a population with a differentiated ancestral genetic pattern (Amerindian component), in addition to considerable data on severe toxicities associated with the treatment of childhood ALL in patients with this genetic profile. Due to this scenario, we have carried out studies to elucidate the genetic profile associated with high toxicities, even without clarification so far, which leads us to investigate different types of markers and even those that do not have references in the literature with the ADME pathways of the ALL treatment. The markers in miRNA genes or genes encoding proteins in the miRNA synthesis complex are different targets of investigation and that is why we chose to include them.
Point 2: In addition, some SNVs, such as ATXN1 (rs68082256) that was linked with neurotoxicity in pediatric ALL, and SLC19A1 (rs1051266) that was significantly associated with MTX-related toxicity in pediatric ALL, were not on the list. In this regard, the rationale for the 25-SNV testing list is unclear. The selection criteria and supporting references need to be further provided.
Response 2: The genetic variants cited by the reviewer were not included in the panel used in this study because they are not located in miRNA genes or genes encoding proteins in the miRNA synthesis complex, which was a selection criterion. Some variants have not yet been described in association with susceptibility or PGx of ALL, however they were added in the study due to their association as markers of susceptibility/carcinogenesis with some types of cancer (XM PAN, 2016; YAJING FENG, 2016; POLTRONIERI-OLIVEIRA, 2017; KAZEMI, 2020; LI YING, 2016). This explanation and broader investigation is very relevant for the discovery of new markers that may share carcinogenic pathways and influence the response to ALL therapy. This information was added to the methodology (lines 240/241).
Point 3: How did the authors define toxicities? The clinical symptoms and signs or the drug plasma concentrations for each type of toxicity (infectious, gastrointestinal, hematologic and neurologic) need to be clarified in detail.
Response 3: Thanks for the suggestion. Toxicities were defined according to the Common Terminology Criteria for Adverse Events V.5. The description of this classification was added in the methodology (lines 234-236), including the corresponding bibliographic reference[9] and highlighted in table 1 (line 81/82). Due to this inclusion, the other references were also reordered.

Reviewer 2 Report
This study evaluated the role of 25 SNVs in microRNA genes and genes encoding proteins of the miRNA SC, in 77 patients treated for ALL-B from the Brazilian Amazon.
I only have a few minor comments that would help improve the manuscript:
1. It should be noted if the genotype distributions were in the Hardy-Weinberg equilibrium
2. Justify why the risk analysis was done under the codominant inheritance model. Prior to the risk analysis, why was the best inheritance model not established for each SNP (codominant, dominant, recessive, overdominant, and additive)?. This observation should be clarified in the manuscript.
3. In supplementary table 1 must be indicated to which name/version of the human genome reference corresponds to the Chromosomal location. In addition, the minor allele frequency must be indicated.
4. Given the size of the sample, the statistical power of the study and its limitations should be indicated.
Author Response
Point 1: It should be noted if the genotype distributions were in the Hardy-Weinberg equilibrium.
Response 1: We appreciate the opportunity to clarify your request. An analysis of the frequencies of the genotypes of each genetic variant was carried out, where it was shown that of the 25 variants investigated, 17 were presented in HWE (p-value > 0.05), as shown in Picture Supplementary 1. The imbalance observed in 8 polymorphisms did not compromise our results, given that the included population has a different genetic profile, with a strong contribution from Amerindian ancestry.
Point 2: Justify why the risk analysis was done under the codominant inheritance model. Prior to the risk analysis, why was the best inheritance model not established for each SNP (codominant, dominant, recessive, overdominant, and additive)?. This observation should be clarified in the manuscript.
Response 2: The exposed codominant model was chosen because it was the one that demonstrated the greatest statistical relevance in our results. Unlike the dominant analysis models, the codominant analysis makes it possible to differentiate homozygous and heterozygous individuals. In addition, with this analysis model, it is possible to infer possible functional associations of these variants, facilitating their reflection on the effect on the manifestation of the disease or clinical event, such as toxicity, whether risk or protection.
Point 3: In supplementary table 1 must be indicated to which name/version of the human genome reference corresponds to the Chromosomal location. In addition, the minor allele frequency must be indicated.
Response 3: We appreciate the suggestion. The references to chromosomal locations were obtained from the Genome Reference Consortium of the human genome reference assembly, GRCh38. This information and the frequency of the smallest allele have been added in the legend of Supplementary Table 1.
Point 4: Given the size of the sample, the statistical power of the study and its limitations should be indicated.
Response 4: Thanks for the suggestion. Acute lymphocytic leukemia (ALL) accounts for about 40% of hematological malignancies in the Amazon region. In the state of Pará, this proportion reaches 45%, with approximately 70 cases diagnosed annually, where 63% of cases are children up to 9 years old (CORREA NETTO et al., 2020). The population of ALL patients included in the present study is representative, although relatively small, totaling 77 patients, but a higher number than the average number of cases diagnosed annually in the State. This shows that the sample number has a significant statistical power, which allows the inference of the evidenced results. In addition, it is a peculiar population, composed of a highly mixed population, with a strong Amerindian contribution, which differs from the rest of the world.
Reference: Correa-Netto, N.F. Martins, D.P. Melo, N. Loggetto, S.R. Panorama do atendimento ambulatorial e hospitalar dos pacientes diagnosticados com leucemia no Brasil: uma análise quantitativa. Hematol transfus cell ther.2020; 42(S2):S1–S567. doi: https://doi.org/10.1016/j.htct.2020.10.291.

Round 2
Reviewer 1 Report
The authors have properly addressed my questions and comments.